# Silicic Acid Polymerization and SiO_2_ Nanoparticle Growth in Hydrothermal Solution

**DOI:** 10.3390/polym14194044

**Published:** 2022-09-27

**Authors:** Vadim V. Potapov, Angel A. Cerdan, Denis S. Gorev

**Affiliations:** 1Research Geotechnological Center, Far Eastern Branch of Russian Academy of Sciences, 30, Severo-Vostochny Highway, 683002 Petropavlovsk-Kamchatsky, Russia; 2Chemical Department of Moscow State University, Leader Scientist, Russian Federation, Leninskie Gory, 1, 119991 Moscow, Russia

**Keywords:** hydrothermal synthesis of SiO_2_ nanoparticles, kinetics of orthosilicic acid polymerization, nanoparticle size distribution

## Abstract

The approach of numerical simulation of orthosilicic acid OSA polymerization and SiO_2_ nanoparticle formation in hydrothermal solution have been developed based on the model of the homogeneous stage of nucleation and the subsequent growth of particles. The influence of surface tension on the interface of SiO_2_–water, the rate of molecular deposition, and Zeldovich factor Z were evaluated. Temperature dependence on time, pH, initial OSA concentration, and ionic strength are the main parameters that determine the kinetics of colloid phase formation, the final average size of SiO_2_ nanoparticles, and the particle size distribution and its polydispersity index. The results of the numerical simulation were verified with experimental data on OSA polymerization and measurement of nanoparticles sizes using the method of dynamic light scattering in a wide range of temperatures of 20–180 °C, pH = 3–9, SiO_2_ content C_t_ of 300–1400 mg/kg, and ionic strength I_s_ of 0.0001–0.42 mol/kg. The results obtained can be used in the technology of hydrothermal synthesis of sols, gels, and nanopowders to regulate the kinetics of OSA polymerization and SiO_2_ nanoparticle growth, particle size distribution, morphology, and structure of products.

## 1. Introduction

Hydrothermal synthesis of SiO_2_ nanoparticles is the new technological method with a low production cost, high density of silanol groups, low toxicity, excluding the use of chemical reagents and other green chemistry methods of nanoparticle synthesis [1,2,3], and allows to obtain different forms of nanosilica—sols, gels, powders, products of the sol-gel process applied in concrete, agricultural plants, medicines, and other materials [4,5,6,7]. The technology of hydrothermal synthesis includes the stages of OSA polymerization, ultrafiltration membrane concentration of nanoparticles, the sol-gel process, vacuum sublimation, and others [4,5,6]. The temperature regime, the stage of polymerization, pH, initial OSA concentration C_s_, and ionic strength influenced the rate of SiO_2_ nanoparticle formation, the average nanoparticle size, and the index of polydispersity of nanoparticles in hydrothermal sols. Therefore, the density, morphology, agglomerate dimensions and fractal structures, volume, surface, diameters, and structure of pores of nanopowders produced from hydrothermal sols are also affected [4,5]. There is a need to regulate the parameters at the stage of OSA polymerization of silicic acid by means of numerical simulation to achieve the necessary characteristics of sols, gels, and nanopowders.

The kinetics of OSA polymerization with different mechanisms has been studied by many researchers under various conditions: pH, initial concentration of dissolved silica Cs, solution temperature, ionic strength, and sizes and molecular weight of SiO_2_ particles formed.

At sufficiently high concentrations of silica in 2–6 wt.% sodium silicate salt solutions with the ratio SiO_2_:Na_2_O = 3.25 and with acidification at pH = 1.7, when the increase in molecular weight was accompanied mainly by the aggregation of small particles the degree of polymerization increased proportionally to the square root of the polymerization time [8].

At the monomer concentration of 0.24 % in the absence of salts [9,10], the average molecular weight at pH = 2.0 increased linearly depending on the square root of time; at pH = 3.2, 3.8, it increased proportionally with time; at pH = 4.36—linearly from the square of time. At pH = 2.2, temperature 90 °C, the average particle diameter was 1.3 nm. At a higher silica concentration of 8 g/L after about 150 h at pH = 3.0 or 75 h at pH = 2.0 the average molecular weight reached 300 SiO_2_ with a smaller resulting particle size of 0.9 nm [11]. The average molecular weight of the newly formed polymerized acid prepared by removing Na from a solution of sodium silicate at pH= 2–3 was about 660 SiO_2_ atoms [12]. At pH = 2.1, temperature 25 °C, in 0.24 wt.% SiO_2_ solution after 60 days after the start of polymerization, the proportion of monomeric silica was 14% and the particle diameter was 1.4 nm [13]. At pH = 8.8, growth of particles occurred during 1 min, that is 3∙10^4^ times faster than at pH = 2.0. In studies performed at 25 °C, pH = 2.0, in the solution of monosilicic acid 2.4% SiO_2_ for 24 days the particle diameter was 1.65 nm; after 8 days it was about 3.4 nm [14,15,16]. In a wide range of SiO_2_ contents at pH =3.0 and pH = 6.1, the initial average diameter of the nuclei was 1.5 nm and the final particle size was about 3 nm [17]. In [18], after 30 days of polymerization at pH = 7.0 and 25 °C with a total concentration of SiO_2_ in a solution of 0.4 g/L the diameter of nanoparticles corresponding to the solubility was 3.7 nm.

In the pH = 7.0–10.0 region, when colloid particles are formed, but the aggregation process cannot proceed, the reaction order was 3 [13,19]. However, in [20], at different initial concentrations of OSA the apparent “order” of the polymerization reaction was determined as equal to 2 at a pH above 2 and was equal to 3 at a pH below 2 [20], whereas the second-order reaction rate constant decreased.

It was found that the rate of polymerization was the greatest at pH = 8.3 and close to the maximum in the pH = 7.0–9.0 region, and at pH < 7.0 and pH > 9.0 it noticeably decreased [21].

The detailed study of polymerization in the wide area of concentrations of 0.2–1.8 g/L of silica and pH = 4.0–10.0 at 25 °C confirmed that there is an induction period during which the polymerization of the monomer is only slight or nonexistent. A decrease in the monomer content as it polymerizes at pH = 7.0 and 25 °C at an initial content of 0.05 wt. % occurred according to the law 0.37∙(t − 0.35)^−0.333^ [22]. The induction period was 0.35 h. After 1 h after the decrease in the monomer content, silica particles began to grow rapidly without the formation of any intermediate product. The presence of a long induction period at pH = 4.5–5.5 and a temperature of 95 °C is shown in [23]. The induction period characterizes the metastability of the system and for aqueous silicic acid at pH = 7.75, t = 20 °C, C_s_ = 1450 mg/kg under mixing conditions it decreased from 3.25 to 2.75 h [20,21]. In the strongly acidic region, the rate of the process is proportional to [H+]^1.2^ at pH < 1.8, and in the weakly acidic region at pH > 3.4 –[OH-]^0.9^ [24,25]. The rate increases initially with an increase in temperature from 20 to 90 °C and decreases with a decrease in the initial OSA concentration [24,25]. The results obtained in [26] for OSA solutions with a SiO_2_ content of 300–1200 mg/kg and low salinity in the temperature range from 5 to 180 °C confirmed that the temperature changes at a constant silica concentration do not significantly affect the increase in induction time at temperatures from 90 to 180 °C.

In a dimer–monomer mixture at pH = 2.0 [27] the polymerization mechanism and particle sizes differed from the results of monosilicic acid polymerization obtained for polycyclic polymer formation in [14,15,16]: at the beginning the discrete particles of less than 4 nm in size formed, then such particles aggregated [27].

Thus, the experimental data obtained by the authors showed significant differences in the values of the polymerization reaction order, the rate constant, the induction period, the sizes of the formed nanoparticles, and the different effects of temperature, pH, and ionic strength on these parameters. Taking this into account, it is justified to use the method of numerical simulation of polymerization based on a certain mathematical model.

One of the mathematical models of orthosilicic acid polymerization is Fleming’s model [28]. Fleming suggested that the process under study is characterized by two main areas of flow. In the first region, the initial concentration of silicic acid exceeds the pseudo-equilibrium concentration of Cx, and the polycondensation reaction is a first-order reaction both with respect to the difference (C_s_-C_x_) and with respect to the surface concentration of C_SiO_-ionized groups of SiO^−^, that is, the surface charge of colloid particles. The second area is defined by the condition C_s_ < C_x_.

The decrease in the rate of transition of monomeric silica to colloid is explained using the Weres-Yi-Tsao model [29]. The authors studied two processes: the formation of colloid particles at the homogeneous stage of nucleation and the subsequent growth of particles due to OSA molecular deposition. From this model, developed on the basis of classical concepts of nucleation by Lose-Pound, it follows that the supersaturation of S_N_ (T), equal to C_s_/C_e_, C_e_ (solubility of amorphous silica in water at temperature T), and pH are the main factors determining the rate of nucleation I_N_ of silicic acid in an aqueous solution.

The authors of [30] used two methods to study the OSA polymerization, transfer in nanocolloids, and nanosilica precipitation in synthetic solutions based on the Na_2_SiO_3_ precursor: (1) the concentration model, when the rate of monomeric silica concentration C_s_ decrease is proportional to the 4-th order of k_1_∙C_s_^4^ and the rate of nanosilica transfer to precipitated silica is proportional to the 1-st order of k_2_∙C_nano_; and (2) the supersaturated model, when the rate of monomeric silica concentration C_s_ decrease is proportional to the 4-th order of the difference k_1_∙(C_s–_C_e_)^4^, C_e_—solubility of amorphous silica in the water at 25 °C. In the wide range of initial C_s_ 240–1200 mg/kg, ionic strength 0.01–0.24 mol/kg, and pH = 3–7 it was obtained that constants k_1_ and k_2_ did not differ for concentration and supersaturation models. The models cannot predict the sizes and distribution of SiO_2_ particles.

The method of molecular dynamic simulation for OSA polymerization was developed in [31,32] at the temperature up 2700 K and can provide the structure of polymerized clusters which was in accordance with the ^29^Si NMR method.

The models with different orders of polymerization reaction [8], taking into account the concentration of OH^−^ groups [33] and ionic strength [28] have been used for analysis of experimental results of orthosilicic acid polymerization in hydrothermal solutions [34].

In [35], the model of Weres-Yi-Tsao [29] has been applied to experimental results in hydrothermal solutions because it can predict not only time dependence of the OSA concentration but also evaluate how the average size of SiO_2_ nanoparticles depends on time. The main purpose of this work was to investigate nanoparticle size distribution dependence on the temperature, ionic strength, and other parameters of the polymerization process needed to regulate the characteristics of nanosilica sol, gel, and powders produced using the technology of hydrothermal synthesis [4,5,6,7]. For this purpose, it is necessary: (1) to study the dependence of surface tension σ_sw_, rate of molecular deposition R_md_, and Zeldovich factor Z on the temperature and pH to explain the mechanisms of their influence on the results of the polymerization process; (2) to examine the dependence of the average nanoparticle size and induction period on temperature, pH, and ionic strength; (3) to study the nanoparticle size distribution under conditions of time variable temperatures; (4) to verify the results of the numerical simulation with the experimental data in wide range of T, pH, and I_s_; and (5) to compare the results of the numerical simulation under time-variable temperature conditions at different technological stages of the hydrothermal sol production with the results of determining the average size of SiO_2_ nanoparticles and their size distribution using the DLS method.

## 2. Method of Numerical Simulation of OSA Polymerization

The initial concentration of molecules of orthosilicic acid formed due to dissolution of minerals of rocks [4,5,6,7] is determined using quartz solubility C_cr_ (mol/kg) of α-quartz in water at the temperature T(K) [36]:(1)$${\mathrm{lgC}}_{\mathrm{cr}}=-1.468+\frac{252.9}{T}-\frac{3.217\cdot 10^{5}}{T^{2}}.\;$$

After the hydrothermal solution reaches the surface pressure and the temperature decreases, the supersaturation of the solution depends on solubility C_e_ of amorphous silica [37]:(2)$${\mathrm{lgC}}_{e}=-0.1185-\frac{1.126\cdot 10^{3}}{T}+\frac{2.3305\cdot 10^{5}}{T^{2}}–\frac{3.6784\cdot 10^{7}}{T^{3}}.\;$$

Polymerization of orthosilicic molecules in oversaturated solution is provided by the formation of siloxane bonds and dehydration:(3)$$H_{4}{\mathrm{SiO}}_{4}+H_{4}{\mathrm{SiO}}_{4}\;\to \;{\mathrm{Si}}_{2}O_{7}H_{6}+H_{2}O$$

The total silica concentration C_t_ in the solution is equal to the sum of the concentrations of colloid silica C_col_, soluble silicic acid C_s_, and ions of silicic acids C_in_:(4)$$C_{t}=C_{\mathrm{col}}+C_{s}+C_{\mathrm{in}}.\;$$

The fraction of dimers does not exceed 1.0% from C_e_, the fraction of trimers is no more than 0.1% [26], the fraction of tetramers and low-molecular cyclic polymers is less than 0.1%, and the fraction of C_in_ does not exceed 14.0%.

The physical and chemical characteristics of the hydrothermal solution from the Mutnovsky geothermal electric plant are presented in the Table 1.

We developed the mathematical model [29] which is based on the calculation of the rate of nucleation I_N_ (nucl/(kg∙s)) dependent on oversaturation S_N_ (T) = C_s_/C_e_: (5)$$I_{N}=Q_{\mathrm{LP}}\cdot Z\cdot \left(R_{\mathrm{md}}\cdot A_{\mathrm{cr}}\cdot N_{A}\cdot \rho \cdot M_{\mathrm{Si}}^{-1}\right)\cdot e^{-\frac{\Delta \mathrm{Fcr}}{k_{B}\cdot T}}.\;$$
where ΔF_cr_ is the change in free energy due to the formation of a nucleus of critical radius R_cr_ with a surface area of A_cr_ = 4·π·R_cr_^2^ and the number of SiO_2_ molecules n_cr,_ σ_sw_ is the surface tension at the silica–water interface, R_md_ is the rate of molecular deposition of SiO_2_ on a solid surface, kg/(m^2^∙s), k_B_ = 1.38∙10^−23^ J/K, M_Si_ = 0.060 kg/mol, N_A_ =6.02∙10^23^ mol^−1^, Q_LP_ is the Lothe–Pound factor, Q_LP_ = 3.34∙10^25^ kg^−1^, and Z is the Zeldovich factor, calculated as follows:(6)$$R_{\mathrm{cr}}=2\cdot \sigma_{\mathrm{SW}}\cdot \frac{M_{\mathrm{Si}}}{\rho \cdot N_{A}\cdot k_{B}\cdot T\cdot {\mathrm{lnS}}_{N}}.\;$$
(7)$$n_{\mathrm{cr}}=\left(\frac{4\pi }{3}\right)\cdot \left(\frac{\rho \cdot N_{A}}{M_{\mathrm{Si}}}\right)\cdot R_{\mathrm{cr}}^{3}.\;$$
(8)$$\Delta F_{\mathrm{cr}}=\frac{\sigma_{\mathrm{SW}}\cdot A_{\mathrm{cr}}}{3}={\left(\frac{16\cdot \pi }{3}\cdot \sigma_{\mathrm{sw}}^{3}\cdot \frac{M_{\mathrm{Si}}}{\rho \cdot N_{A}\cdot k_{B}\cdot T\cdot {\mathrm{lnS}}_{N}}\right)}^{2}\;.\;$$
(9)$$Z={\left[-\left(\partial^{2}\Delta F_{\mathrm{cr}}/\partial n_{\mathrm{cr}}^{2}\right)/\left(2\cdot \pi \cdot k_{B}\cdot T\right)\right]}^{0.5}=\left(\frac{2}{3}\right)\cdot {\left(\frac{3\cdot M_{\mathrm{Si}}}{4\cdot \pi \cdot \rho \cdot N_{A}\cdot n_{\mathrm{cr}}^{2}}\right)}^{\frac{1}{3}}\cdot {\left(\frac{{\;\sigma }_{\mathrm{sw}}}{k_{B}\cdot T}\right)}^{0.5}.\;$$
with ρ = 2200 kg/m^3^—the density of amorphous silica.

The function R_md_, which is the rate of molecular deposition of SiO_2_ as a function of temperature and pH of the solution, is expressed as follows in the model [29]:(10)$$R_{\mathrm{md}}=F\left(\mathrm{pH},{\;\mathrm{pH}}_{\mathrm{nom}}\right)\cdot k_{\mathrm{OH}}\left(T\right)\cdot f_{f}\left(S_{a}\right)\cdot \left(1-S_{N}^{-1}\right).\;$$
(11)$$\mathrm{lg}k_{\mathrm{oH}}=3.11-\frac{4296.6}{T}.$$
(12)$$f_{f}=S_{t}^{5},{\;S}_{a}<S_{t}.$$
(13)$$f_{f}=S_{t}^{5}+5\cdot S_{t}^{4}\cdot \left(S_{a}-S_{t}\right),{\;S}_{a}>S_{t}.\;$$
(14)$$\mathrm{lg}S_{t}=0.0977+\frac{75.84}{T}.\;$$
(15)$$F\left(\mathrm{pH},{\mathrm{pH}}_{\mathrm{nom}}\right)=h_{f}\cdot f^{\prime }\left(\mathrm{pH}\right)+\left(1-h_{f}\right)\cdot f^{\prime }\left({\mathrm{pH}}_{\mathrm{nom}}\right).\;$$
(16)$$f^{\prime }\left({\mathrm{pH}}_{\mathrm{nom}}\right)=\frac{f\left({\mathrm{pH}}_{\mathrm{nom}}\right)}{f\left(7.0\right)}.\;$$
(17)$$\mathrm{lg}f=\mathrm{pH}-{\mathrm{pK}}_{i}+\mathrm{lg}\left[{\mathrm{Na}}^{+}\right].$$


The coefficient of surface tension σ_sw_ depends on temperature and pH via the function I(pH, pH_nom_):
(18)$$\sigma_{\mathrm{sw}}=H_{\sigma }-T\cdot S_{\sigma }-2.302\cdot 10^{-3}\cdot n_{0}\cdot k_{B}\cdot T\cdot I(\mathrm{pH},{\mathrm{pH}}_{\mathrm{nom}}).$$
(19)$$I\left(\mathrm{pH},{\;\mathrm{ph}}_{\mathrm{nom}}\right)=0.119\int_{-\infty }^{\mathrm{pH}}F\left({\mathrm{pH}}^{\prime },{\mathrm{pH}}_{\mathrm{nom}}\left({\mathrm{pH}}^{\prime }\right)\right)\mathrm{dp}H^{\prime }.\;$$
where S_a_ = (1 − α_i_)·S_N_, α_i_ is a fraction of silicic acid in ionic form, pH_nom_ = pH + lg([Na^+^]/0.069), [Na^+^] is an ion activity [Na^+^], mol/kg, pK_i_ = 6.4, f(7.0) = 0.119, h_f_ = 0.45, H_σ_ and S_σ_ are specific enthalpy and entropy of silica surface in water, H_σ_ = 63.68·10^−3^ J/m^2^, S_σ_ = 0.049·10^−3^ J/m^2^·K, and n_o_ = 6.84∙10^18^ m^−2^ [29].

Time dependence I_N_ (t) was calculated considering the induction time τ_in_ that is necessary to grow and form a stable population of the particles with sizes close to critical [29]:(20)$$I_{N}\left(t\right)=I_{N}\cdot \left(1-e^{-\frac{\mathrm{Tp}}{\tau_{\mathrm{in}}}}\right)\;.\;$$
(21)$$\tau_{\mathrm{in}}=1.08\cdot 10^{-6}\cdot {\left(R_{\mathrm{md}}\right)}^{-1}\cdot {\left(Q_{\mathrm{LP}}\cdot Z\cdot R_{\mathrm{cr}}^{2}\cdot e^{-\frac{\Delta \mathrm{Fcr}}{k_{B}\cdot T}}\right)}^{-0.25}.\;$$

Initial values were entered: the temperature T, pH, concentration of ions and ionic strength of the solution, total silica C_t_ content, time step DT_p_, initial radii R(I), and particle quantity MP(I) of every class “I”. The Runge–Cutt method was used for the numerical simulation.

The total content of colloid, monomeric silica, and the current value of supersaturation were estimated:(22)$$M_{\mathrm{col}}=\sum_{I\;=1}^{I\;=\;N}\frac{4\cdot \pi \cdot \rho \cdot R^{3}\left(I\right)\cdot \mathrm{MP}\left(I\right)}{3}.\;$$
(23)$$C_{s}=C_{t}-M_{\mathrm{col}}.\;$$

Solution degree of supersaturation S_N_ was:(24)$$S_{N}=\frac{C_{s}}{C_{e}\left(T\right)}.\;$$

Then, the calculation of values of σ_sw_, R_md_, R_cr_, Z, τ_in_, and I_N_(t) corresponding to the current values of S_N_, pH, and T using the equations was undertaken.

The quantity of new particles appearing during the time DT_p_ in accordance with the current value of nucleation rate I_N_ on a given program step (N+1) was calculated and it was equal to the quantity of particles in the new class N+1:
(25)$$\mathrm{MP}\left(N+1\right)=I_{N}\cdot {\mathrm{DT}}_{p}$$


The summary of the concentration of particles CONP was calculated in a program cycle:(26a)$$\mathrm{CONP}=\sum_{I\;=1}^{I\;=\;N+1}\mathrm{MP}\left(I\right)\;.\;$$

The increment in mass DPM of every class “I” during the time DT_p_ was calculated:
(26b)$$\mathrm{DPM}\left(I\right)=4\cdot \pi \cdot R^{2}\left(I\right)R_{\mathrm{md}}\cdot {\mathrm{DT}}_{p}$$


The calculation of particle radius R(I) corresponding to a new value of particle mass in every class I was:(27)$$R\left(I\right)={\left(R^{3}\left(I\right)+\frac{3\cdot \mathrm{DPM}\left(I\right)}{4\cdot \pi \cdot \rho }\right)}^{\frac{1}{3}}.\;$$

The mean values of R_a_, R^2^_a_, and R^3^_a_ in all classes of particles were estimated using the program cycle: (28)$$R_{a}=\sum_{I+1}^{I\;=\;N+1}\frac{R\left(I\right)\cdot \mathrm{MP}\left(I\right)}{\mathrm{CONP}}\;.\;$$
(29)$$R_{a}^{2}=\sum_{I+1}^{I\;=\;N+1}\frac{R^{2}\left(I\right)\cdot \mathrm{MP}\left(I\right)}{\mathrm{CONP}}\;.\;$$
(30)$$R_{a}^{3}=\sum_{I+1}^{I\;=\;N+1}\frac{R^{3}\left(I\right)\cdot \mathrm{MP}\left(I\right)}{\mathrm{CONP}}\;.\;$$

## 3. Results

### 3.1. Dependence of the Surface Tension σ_sw_, Rate of Molecular Deposition R_md_, and Zeldovich Factor Z on the Temperature and pH

The main factors influencing the rate of the nucleation process I_N_ and the final average particle size R_a_ are the degree of temperature T, the degree of supersaturation S_N_(t), and pH. In addition, the final result of the numerical simulation is influenced by the functions f(pH, pHnom) and i(pH, pHnom), specified in the Equations (10)–(19). The results of the calculation of the function f(pH) at different pH in the range from 4.0 to 8.7 are presented in Table 1.

With an increase in pH, the value of the f(pH) function increases (Table 2). Consequently, with an increase in pH, the rate of molecular deposition of R_md_ increases, and the R_md_ parameter will strengthen the tendency to decrease the final average particle size. With an increase in the initial concentration of C_s_, the rate of molecular deposition also increases (Figure 1 and Figure 2). An increase in temperature has an inhomogeneous effect on the rate of molecular deposition (Figure 1 and Figure 2): first, due to the increase in the kinetics of the process with increasing temperature, the rate of molecular deposition increases, then R_md_ decreases, since with increasing temperature, the solubility of C_e_ increases, the supersaturation of S_N_ decreases, and the value of the rate of molecular deposition decreases. At a fixed pH, R_md_ reaches a maximum at a certain temperature and the value of this maximum increases with an increase in the initial concentration (Figure 2).

An increase in the rate of molecular deposition with an increase in C_s_ and temperature will contribute to an increase in the rate of nucleation I_N_ of OSA and a decrease in the final average particle size.

The values of the function i (pH) in the range from 6.0 to 8.7 are presented in Table 3. With an increase in pH, the value of the function i (pH) increases. Consequently, the surface tension coefficient and the critical radius R_cr_ reduced. The surface tension coefficient σ_sw_ decreases with increasing temperature and with increasing pH in accordance with Equations (18) and (19) (Figure 3a,b). Consequently, σ_sw_ will be a factor that, with an increase in temperature and pH, will contribute to a decrease in the critical radius R_cr_, an increase in the nucleation rate I_N_, and a decrease in the final average particle size R_a_, competing with the factor of reducing supersaturation S_N_.

The physical meaning of the Zeldovich factor Z, calculated using Equation (9), is that only a certain proportion of particles whose sizes have reached a critical radius continue to grow further. The remaining particles decrease in size. At the same time, the concentration of particles whose sizes are higher than the critical radius is less than the equilibrium concentration. With an increase in temperature at a constant pH, the value of the Zeldovich factor decreases, and at the same time its effect on the rate of nucleation I_N_ competes with the opposite effect associated with a decrease in the supersaturation of S_N_. An increase in pH leads to an increase in the values of the Zeldovich factor (Table 4).

### 3.2. Results of the Numerical Simulation at Different Temperatures and pH: Average Particle Size, Induction Period, and the Particle Size Distribution

The results of numerical simulation showed (Table 5) that when the temperature increases at a constant pH the final average size of SiO_2_ nanoparticles increases because the degree of oversaturation and I_N_ become higher. When the pH decreases at a constant temperature the final average size increases (Table 5) due to the increase in the surface tension and the increase in the critical size of the nucleus R_cr_ and the decrease in I_N_. If the initial concentration of orthosilicic acid increases the final average size decreases because the degree of oversaturation increases, I_N_ increases, and R_cr_ decreases (Table 6).

After nucleation and polymerization completion the fraction of dimers is not more than 1.0 % of solubility C_e_(T), the concentration of trimers is about 0.1 %, and the concentration of tetramers and polymers with circular structures of less than 6 units of SiO_2_ is lower than 0.1 % [26].

The concentration of ions H_3_SiO_4_^−^ and H_2_SiO_4_^2−^ is not more than 14.0 % calculated from the dependences of constants of orthosilicic acid ionization of the first K_1_ and second K_2_ stages:(31)$$\mathrm{lg}K_{1}=-\frac{2549}{T}-15.36\cdot 10^{-6}\cdot T^{2}\;$$
(32)$$\mathrm{lg}K_{2}=5.37-\frac{3320}{T}-20\cdot 10^{-3}\cdot T\;$$

The function of oversaturation S_N_ vs. time T_p_ showed the long period of induction τ_in_ and homogeneous nucleation which becomes much longer at low values of pH when S_N_ changes slightly and the period of rapid growth of SiO_2_ nanoparticles when S_N_ decreases sharply (Figure 4 and Figure 5, Table 7). The number of new particles PNS depending on the duration of nucleation T_p_ reached a constant value when the rate of nucleation R_nuc_ rapidly decreased (Figure 4 and Figure 5). The period of homogeneous nucleation increases when pH decreases and the initial degree of oversaturation S_N_ decreases at higher temperatures and low initial C_s_ concentrations (Table 7). The relation (t_hom_/t_hetg_) increases when the temperature increases and pH decreases (Table 7).

In Figure 6, the distributions of the number of particles vs. radius formed at constant T = 50–60 °C, pH = 4 and 8, and initial OSA concentrations C_s_ = 700 and 800 mg/kg are shown. The form of the function of SiO_2_ particle number distribution by radius has a maximum corresponding to the average radius R_a_. Figure 6 shows an asymmetrical distribution, that is the right branch of the silica particle size distribution formed at constant temperature is convex and the left branch is concave.

The average radius of SiO_2_ nanoparticles and the distribution by radius were determined using the method of dynamic light scattering (DLS). Values R_a_ were in the range from 5.0 up to 20.0 nm and values of particles radii were in the range from 1.0 up to 50.0 nm [4,5,6,7].

### 3.3. Simulation of OSA Polymerization at Different Ionic Strengths

The dependence of C_s_ (T_p_) is also affected by the ionic strength of the I_s_ solution. The simulation results were obtained at t = 20 °C, pH = 8.0, and C_s_ = 700 mg/kg and various values of I_s_: 0.014, 0.07, 0.14, 0.28, 0.4, 0.8, 1.0, 1.2, 1.4, 7.1, 10.6, and 14.2 mol/kg. With an increase in I_s_ from 0.014 to 0.8 mol/kg, the C_s_(T_p_) dependence shifts to the left and acquires a concave shape (Figure 7a,b). In the range of ionic strength values from 0.8 to 1.4 mol/kg, with an increase in I_s_, the rate of decrease of C_s_(Tp) was lower and the curves shift to the right. The shape of the curves becomes convex. At values of I_s_ > 1.4 mol/kg, the curves C_s_ (T_p_) are shifted to the left again.

The dependence of the final average particle size on the ionic strength of the I_s_ solution is non-monotonic also (Table 8, Figure 8). With an increase in I_s_ from 0.014 to 0.8 mol/kg, R_a_ increases. Starting from the value I_s_ = 0.8 mol/kg, the final average particle size decreases. This non-monotonic dependence is explained by the fact that with an increase in I_s_, the rate of molecular deposition of R_md_ increases due to an increase in the degree of ionization of orthosilicic acid α_SIL_. Therefore, the rate of nucleation of OSA I_N_ increases (Equation (5)). At the same time, the growth rate of the formed particles increases (Equation (10)). Thus, the final average size of silica particles is influenced by two competing factors—growth of I_N_ and R_md_ which determines the non-monotonic behavior of the R_a_ (I_s_) dependence.

### 3.4. Simulation of OSA Polymerization under Time-Variable Temperatures

A significant influence of the variable temperature regime on the rate of the OSA nucleation process and the final average size of silica particles was revealed.

To simulate the process of nucleation of OSA under variable temperature conditions, the following law of temperature t reduction with polymerization time T_p_ was chosen:(33)$$t=t_{2}+\left(t_{1}-t_{2}\right)\cdot e^{\frac{-T_{p}}{\mathrm{taun}0}}$$

Dependence (1.34) corresponds to the rate of heat q, W/(m^2^∙s) exchange with an environment with a constant temperature t_o_ according to Newton’s law q = α∙F∙(t − t_0_), t_1_, t_2_ are the initial and final temperatures of the solution; α—coefficient of heat exchange W/(m^2^∙s∙°C); F—area of heat exchange, m^2^; taun0—the characteristic cooling time of the solution. Calculations were performed at t_1_ = 100 °C and t_2_ = 20 °C for solutions of the Mutnovsky hydrothermal field at C_s_ = 700 mg/kg and pH = 8.0; taun0 was taken to be 120, 240, 360, and 480 min.

Figure 9 shows that the smaller the tun0, the higher the level of supersaturation of S_N_. Consequently, the rate of OSA nucleation increases. This leads to a decrease in the final average particle size.

The effect of the initial concentration on the rate of the nucleation process and the final average size of silica particles in the variable temperature regime turned out to be ambiguous. The temperature was lowered from 100 to 20 °C, pH = 8.0, and taun0 = 360 min. The initial concentration assumed various constant values—500, 600, 700, 800, and 1000 mg/kg. Figure 10 shows a comparison of the dependencies obtained.

At variable temperatures, the supersaturation functions of time have a non-uniform behavior. The supersaturation of the solution S_N_ = C_s_/C_e_ depends on the solubility of C_e_, which decreases, and faster than the value of the concentration of C_s_.

Figure 9c and Figure 10c show a more symmetrical distribution, that is both branches of the silica particle size distribution are convex in contrast to the case of constant temperatures when the right was convex and the left was concave. Thus, the temperature regime at the stage of polymerization strongly influenced the index of polydispersity of hydrothermal sols and the density, morphology, agglomerate dimensions, volume, surface, diameters, and structures of pores of nanopowders produced from hydrothermal sols [5].

### 3.5. Comparison of the Experimental Data with Results of the Numerical Simulation

Experimental data C_s_(T_p_) obtained in sodium silicate solutions with Na^+^ cations removed by ion exchange and low ionic strength [26] were in agreement with the results of numerical simulation in the wide range of temperatures and initial OSA concentration C_s_ (Figure 11) [35]: t = 30, 90, 120,150, 180 °C, C_s_ = 383, 620, 110, 1200 mg/kg, pH = 7.0, 7.4, 7.85; I_s_ = 0.000132–0.00134 mol/kg.

The comparison of the experimental data on OSA polymerization and the results of the numerical simulation for the solution of the Pauzhetsky hydrothermal field showed that the discrepancy of about 12–14.2% was within the uncertainty of the yellow molybdate method for determining the concentration of dissolved silicic acid for the following conditions [35]: t = 20 °C; C_s_ = 315.6–338 mg/kg; pH = 8.2–8.35; and I_s_ = 0.04284 mol/kg.

The results of the simulation of OSA polymerization at the temperatures of 84–95 °C and high ionic strength for solutions with different chemical compositions showed an agreement with the experimental data obtained at Wairakei and Broadlands hydrothermal fields [35]: t = 84 °C, pH = 8.1, and C_s_ = 570 mg/kg (Wairakei); t = 95 °C, pH = 8.0, and C_s_ = 620 mg/kg (well 11 at Broadlands); t = 95 °C, pH = 7.7, and C_s_ = 900 mg/kg (well 22 at Broadlands); I_s_ = 0.05–0.06 mol/kg.

The results of the simulation of the distribution of the number of SiO_2_ nanoparticles vs. radius for solutions in which OSA polymerized at the temperatures 33–43 °C, pH = 6.7–8.1, C_s_ = 366–546 mg/kg, and I_s_ = 0.043 mol/kg corresponded to the results of particles radii measured using the DLS method [35,38] in the range of 50–225 nm.

The experimental results of [24,25] on the maximum nucleation rate v_max_ determined as (C_s1_ − C_s2_)/(T_p2_ − T_p1_) were in agreement with the results of the numerical simulation in the region of pH = 4–6, in which the model used [35] was applicable (pH = 4–8).

The comparison of the average diameter measured using helium chromatography in a synthetic solution with an initial C_s_ = 880 mg/kg at 20 °C and pH = 7.0 [39] were in good agreement with results of the numerical simulation [35].

The values of the k_1_ constant for concentration and supersaturation models obtained in [30] for synthetic solutions in the wide range of initial C_s_ 240–1200 mg/kg, ionic strength 0.01–0.24 mol/kg, and pH = 3–7 corresponded to the times T_p_ of C_s_ decrease calculated in our numerical simulations for the same conditions.

The experimental results [35] with C_s_(T_p_) curves characterized by k_p_ constants of the OSA reaction of the polymerization at different temperatures 20–50 °C, different pH = 4–8.5, and ionic strengths 0.01–1.1 mol/kg corresponded to the models [8,28,33].

## 4. Simulation the Particle Size Distribution in Hydrothermal Sol Production Technology

The technological flow sheet for the solution of the Mutnovsky geothermal electric power plant GeoPP includes the well, separator, pipe silencer, tank for the solution cooling and aging to develop OSA polymerization, ultrafiltration membrane module, and volumes for concentrated sol, Figure 12a. The temperature decreases with the linear dependence in every element from the well to silencer from 300 to 96 °C at pH = 8.0 and with the dependence (1.33) in the aging tank in which the temperature decreases from 96 to 70 °C. When the duration of aging was 20 h and taun = 11 h, the average SiO_2_ nanoparticle radius calculated using the numerical simulation was 12.5 nm (Figure 12b) and it was 12.9 nm measured using the DLS method (Figure 13). When the duration of aging was 100 h and taun = 15 h, the final average diameters were 68.6 and 66 nm using the DLS method. If pH = 5.0, the temperature decrease during aging of 24 h was from 96 to 23 °C, taun = 12 h, and the calculated diameter was 160 nm and the diameter measured using the DLS method was 154 nm.

The temperature and aging duration can be varied to regulate the average particle diameter from 5 to 160 nm.

For the solution from wells at Cerro Prieto GeoPP with the values C_s_ = 950 mg/kg, pH = 7.3, I_s_ = 0.4227 mol/kg, aging duration 65 min, taun = 60 min, and temperature decrease from 100 to 30 °C [23,24], the final average radius calculated in the present work was R_A_ = 1.32 nm and it was 1.35 nm as a result of the calculation in [29].

For the solutions of the Wairakei GeoPP, New Zealand, at aging temperatures of 70 and 20 °C during 50 h, C_s_ = 490 mg/kg, and I_s_ = 0.05 mol/kg, the calculated average radii were 60.7 and 9.5 nm, and the radii measured using DLS were 60 and 10 nm.

## 5. Conclusions

The mathematical model we adopted allowed us to investigate the influence of various physical and chemical factors in a wide range of values on the rate of polymerization of OSA in a hydrothermal solution and on the SiO_2_ nanoparticle size distribution. The results of the numerical simulation were verified using a comparison with experimental data. The temperature, pH, initial OSA concentration, and ionic strength are the main parameters that determine the kinetics of colloid phase formation, the final average size of SiO_2_ nanoparticles, and the form of the particle size distribution.

The final average radius decreases when the temperature decreases, and the degree of oversaturation becomes higher taking into account the surface tension, rate of molecular deposition, and Z-factor. The average radius increases when the pH decreases due to the increase in the surface tension and the critical size of the nucleus. The average radius decreases when the initial OSA concentration and degree of oversaturation increases.

The function of oversaturation S_N_ depending on the time of polymerization T_p_ showed the long period of induction τ_in_ and homogeneous nucleation which becomes much longer at low values of pH. The period of homogeneous nucleation increases when pH decreases and the initial degree of oversaturation S_N_ decreases at higher temperatures and low initial C_s_ concentrations. The relation of homogeneous and heterogenous durations (t_hom_/t_hetg_) increases when the temperature increases and pH decreases.

The dependence of the curve C_s_ vs. T_p_ and final average particle size R_a_ on the ionic strength I_s_ is non-monotonic. With an increase in I_s_ from 0.014 to 0.8 mol/kg, the kinetics of OSA polymerization accelerate, but R_a_ increases. Starting from the value of I_s_ = 0.8 mol/kg, the final average particle size decreases. This non-monotonic dependence is explained by the fact that with an increase in I_s_, the rate of molecular deposition of R_md_ increases due to an increase in the degree of ionization of orthosilicic acid α_SIL_. Therefore, the rate I_N_ of OSA nucleation increases. At the same time, the growth rate of the formed particles increases. Thus, the final average size of silica particles is influenced by two competing factors—the increases in I_N_ and R_md_ which determine the non-monotonic behavior.

At variable temperatures, the supersaturation functions of time have a non-uniform behavior. The supersaturation of the solution S_N_ = C_s_/C_e_ depends on the solubility of C_e_, which decreases, and faster than the value of the concentration of C_s_. Both branches of the silica particle size distribution are concave in contrast to the case of constant temperatures. The variation in temperatures during the stage of OSA polymerization can influence the index of polydispersity of particles, its distribution regarding sizes, and the homogeneity and symmetrical form of the distribution. The temperature regime at the stage of polymerization highly influenced the index of polydispersity of hydrothermal sols and, thus, the density, morphology, agglomerate dimensions and fractal structures, volume, surface, diameters, and structures of pores of nanopowders produced from hydrothermal sols.

The results obtained can be used for regulating the kinetics of SiO_2_ nanoparticle formations, sizes, the polydispersity of the distribution of sizes, and the concentrations of nanoparticles in the technology of the hydrothermal synthesis of sol, gel, and powder.

The results can be applied to the technology of the production of synthetic water sols based on Na_2_SiO_3_ precursors and the technology of precipitated silica.

## Figures and Tables

**Figure 1 polymers-14-04044-f001:**
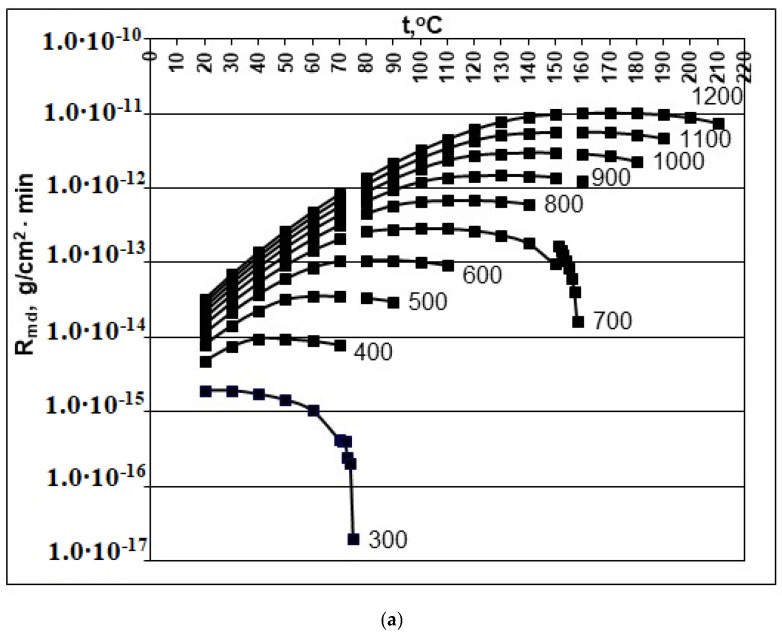
Dependence of the rate of molecular deposition R_md_ (g/(cm^2^∙min)) on the temperature t (°C) and concentration C_s_ (mg/kg). (**a**) pH = 2.0; (**b**) pH = 4.0; (**c**) pH = 6.0; (**d**) pH = 8.0. C_s_ = 300, 400, 500, 600, 700, 800, 900, 100, 110, and 1200 mg/kg.

**Figure 2 polymers-14-04044-f002:**
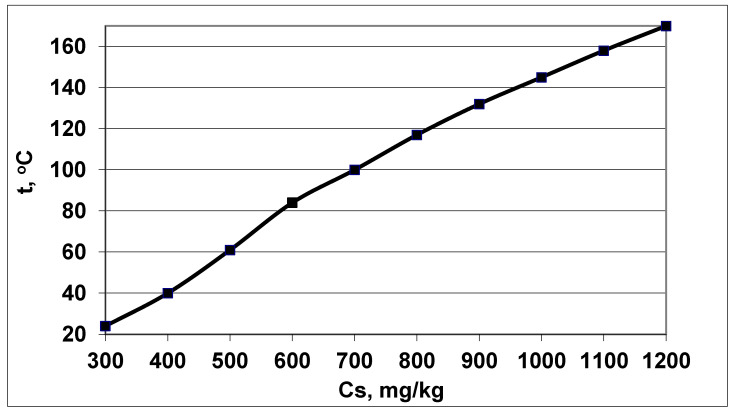
The temperature t (°C) corresponded to the maximum rate of molecular deposition R_md_ at different concentrations C_s_, pH = 8.0.

**Figure 3 polymers-14-04044-f003:**
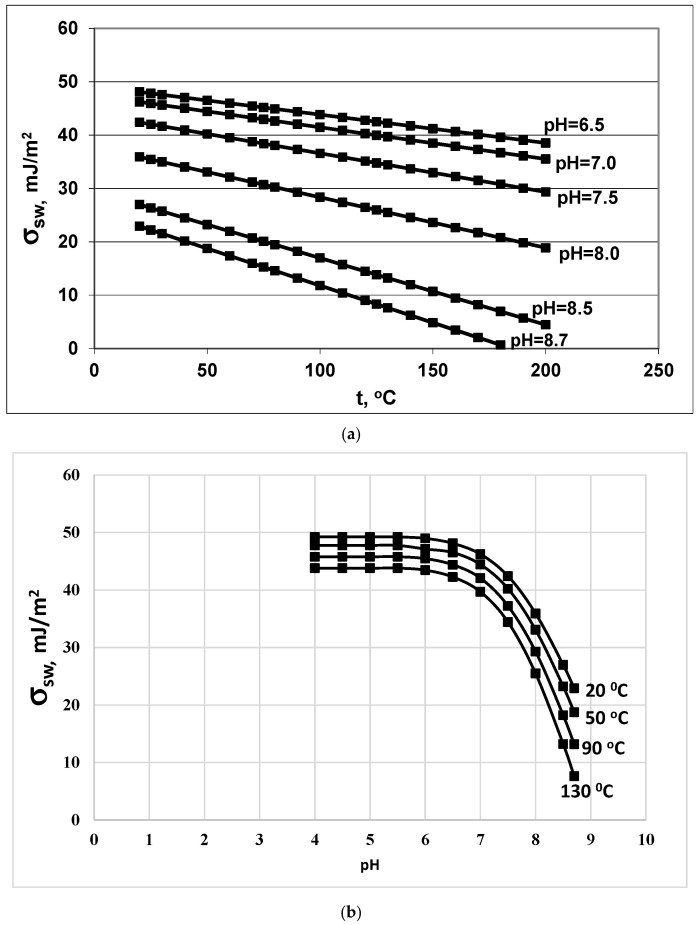
Dependence of the surface tension σ_sw_ (mJ/m^2^) on (**a**) temperature t (°C) and (**b**) on pH. Initial concentration C_s_ = 700 mg/kg.

**Figure 4 polymers-14-04044-f004:**
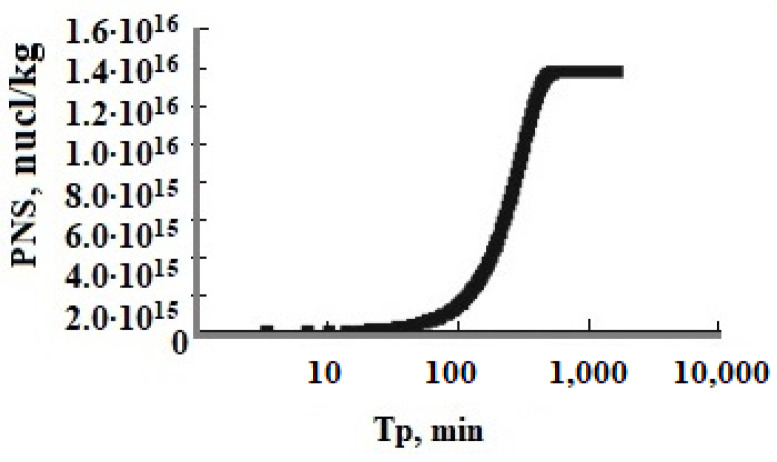
The number of new particles PNS (nucl/kg) versus time T_p_; T = 50 °C, the initial concentration is C_s_ = 700 mg/kg, pH = 8.

**Figure 5 polymers-14-04044-f005:**
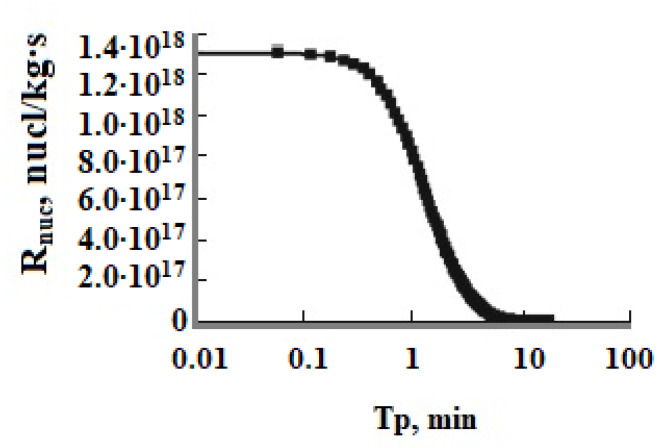
The rate of nucleation R_nuc_ (nucl/(kg∙s)) versus time T_p_; T = 50 °C, the initial concentration is C_s_ = 700 mg/kg, pH = 8.

**Figure 6 polymers-14-04044-f006:**
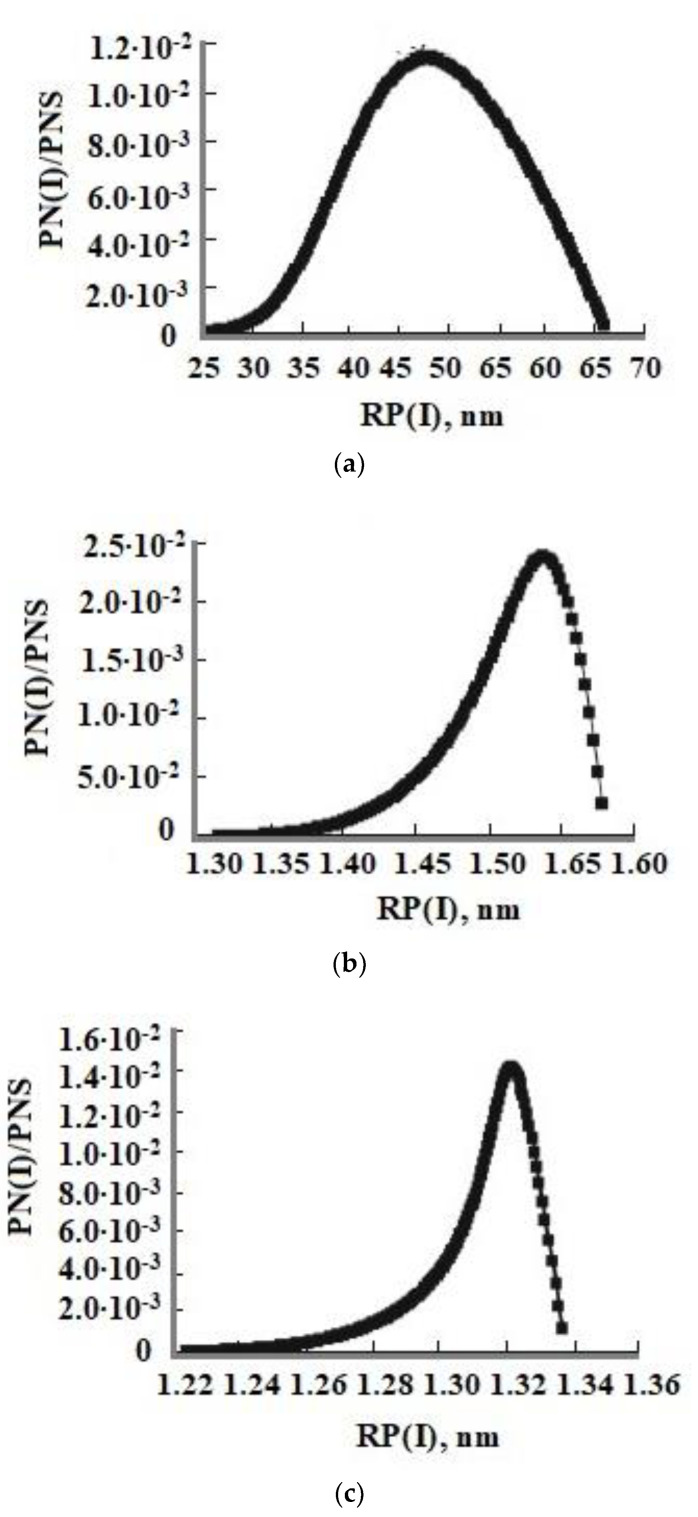
The particle size distribution at different pH and initial OSA concentrations C_s_. (**a**) T = 50 °C, *C*_s_ = 700 mg/kg, pH = 4; (**b**) T = 50 °C, C_s_ = 700 mg/kg, pH = 8; (**c**) T = 60 °C, C_s_ = 800 mg/kg, pH = 8. PNS—total quantity of the formed SiO_2_ particles.

**Figure 7 polymers-14-04044-f007:**
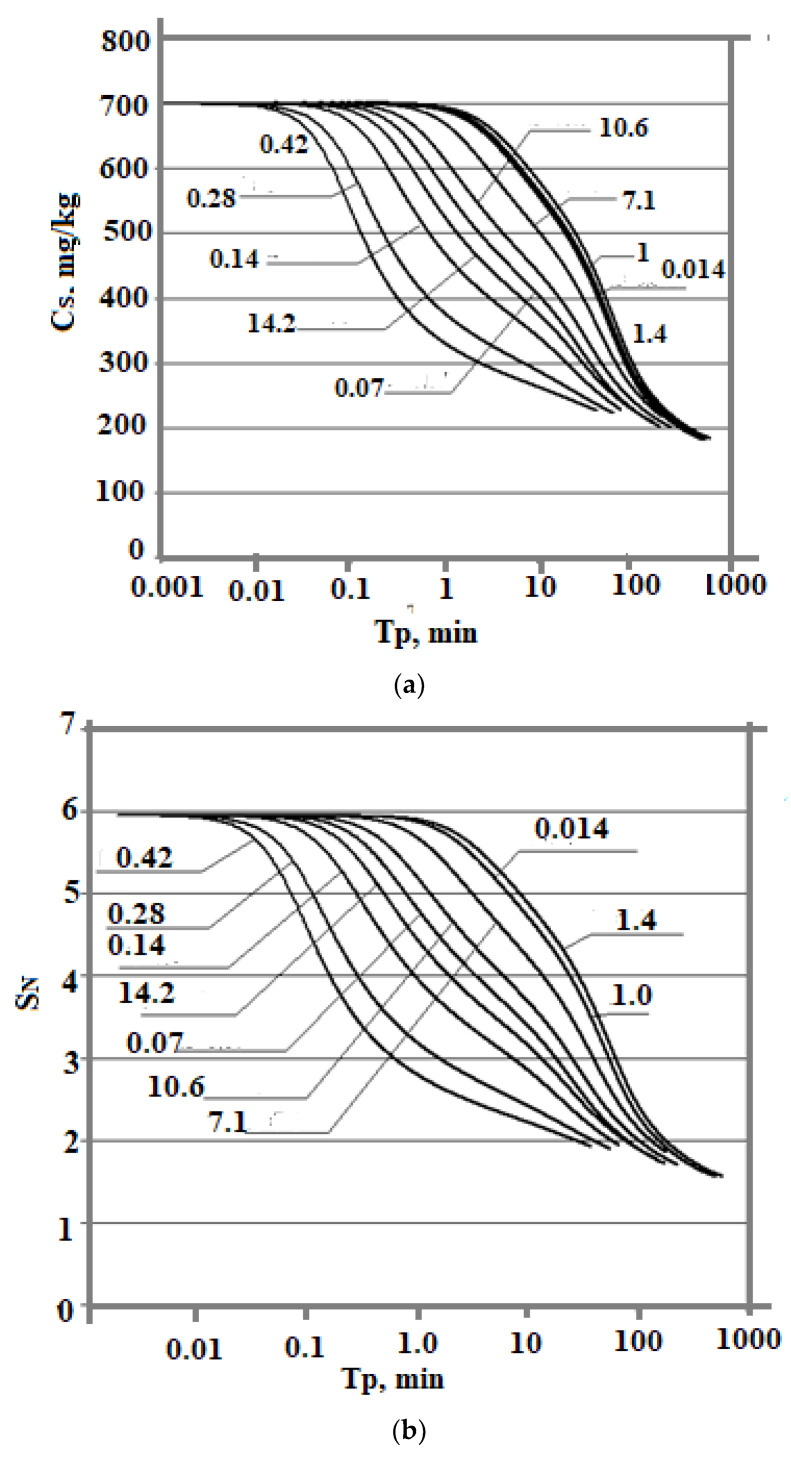
The dependence of the concentration of dissolved silicic acid (**a**) and supersaturation (**b**) on time. t = 20 °C, pH = 8.0, initial concentration C_s_ = 700 mg/kg, and various values of ionic strength (mol/kg).

**Figure 8 polymers-14-04044-f008:**
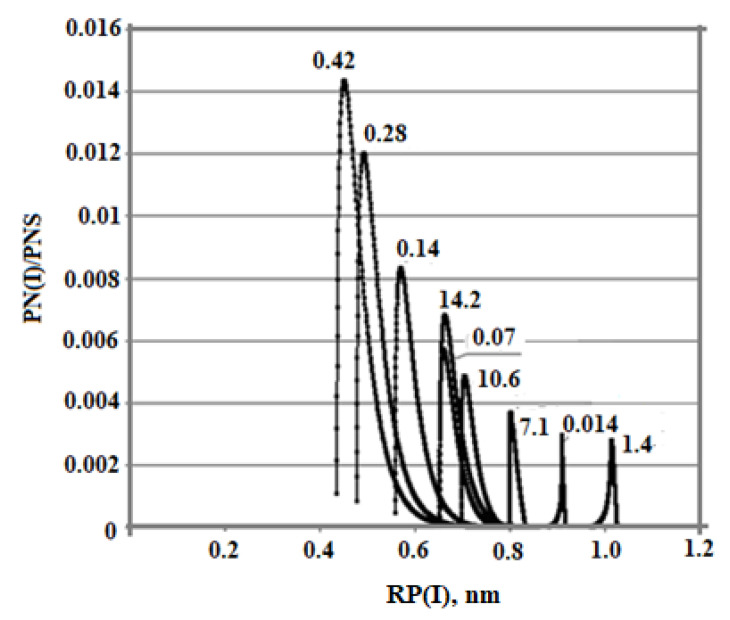
Particle size distribution with different ionic strength values. t = 20 °C, pH = 8.0, and initial concentration C_s_ = 700 mg/kg, I_s_ (mol/kg).

**Figure 9 polymers-14-04044-f009:**
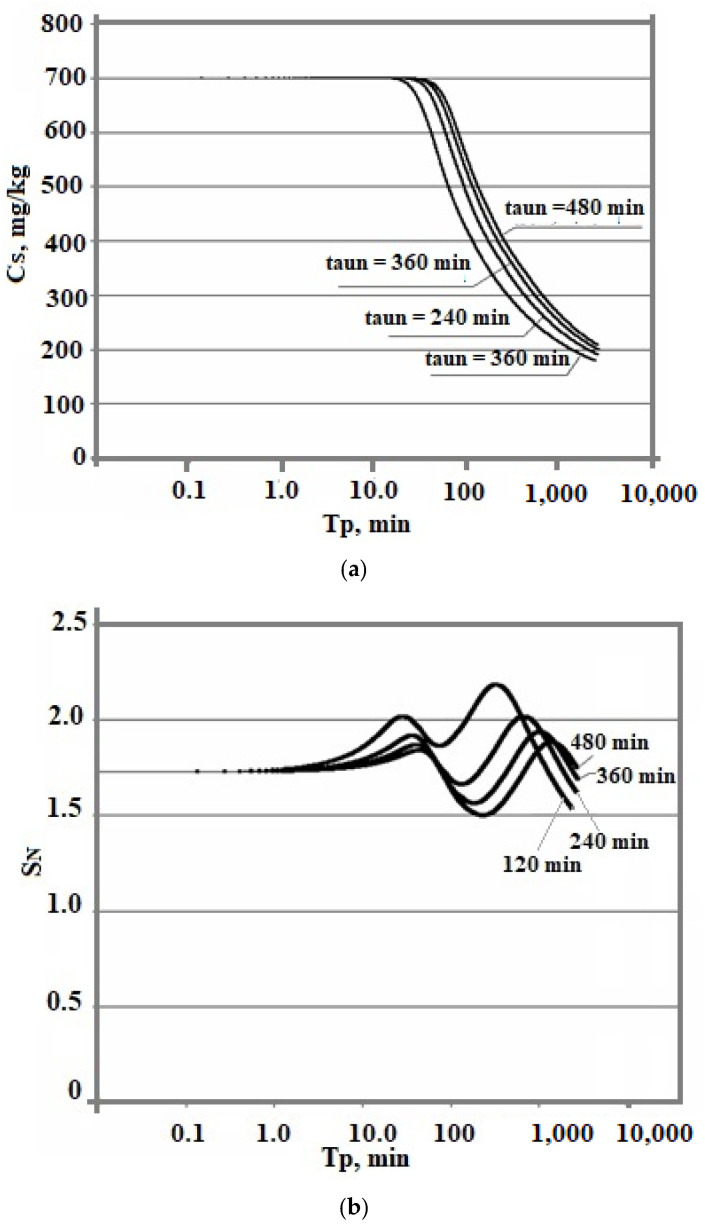
The time dependences of the concentration of dissolved silicic acid C_s_ (**a**), supersaturation (**b**), and particle size distribution (**c**). The temperature decreases from 100 to 20 °C, pH = 8.0, and C_s_ = 700 mg/kg.

**Figure 10 polymers-14-04044-f010:**
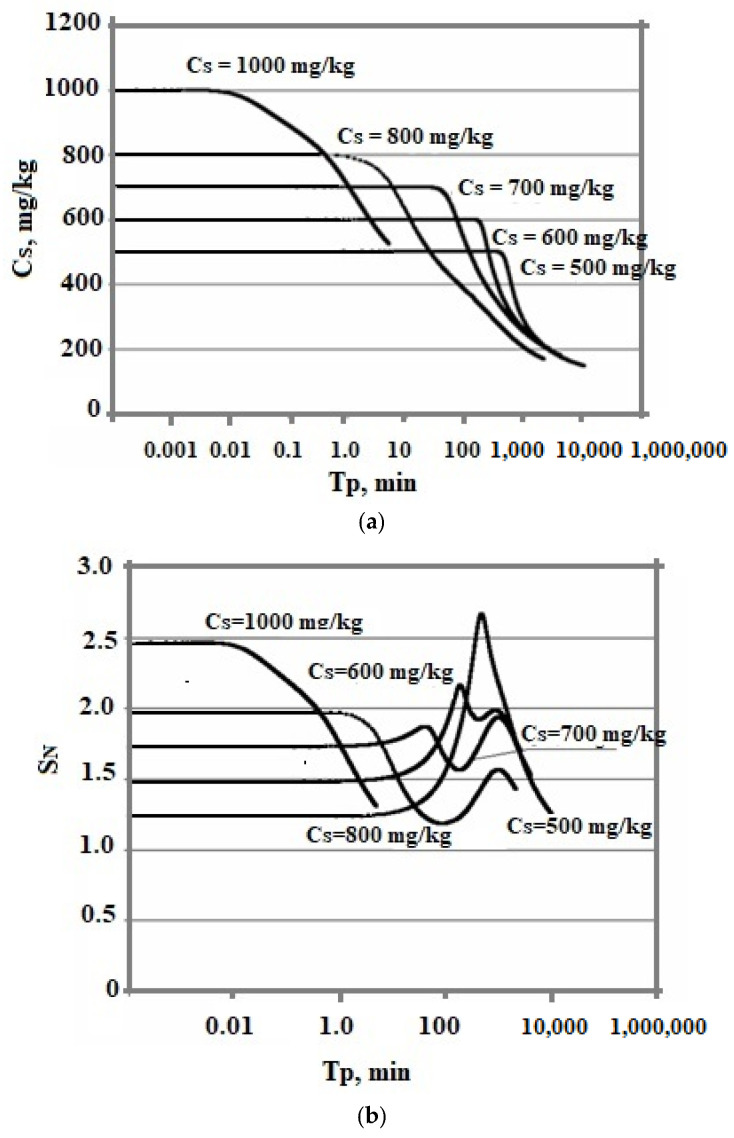
Dependence of C_s_ on T_p_ (**a**), S_N_ on T_p_ (**b**), and particle size distribution (**c**). The temperature decreases from 100 to 20 °C and taun0 = 360 min.

**Figure 11 polymers-14-04044-f011:**
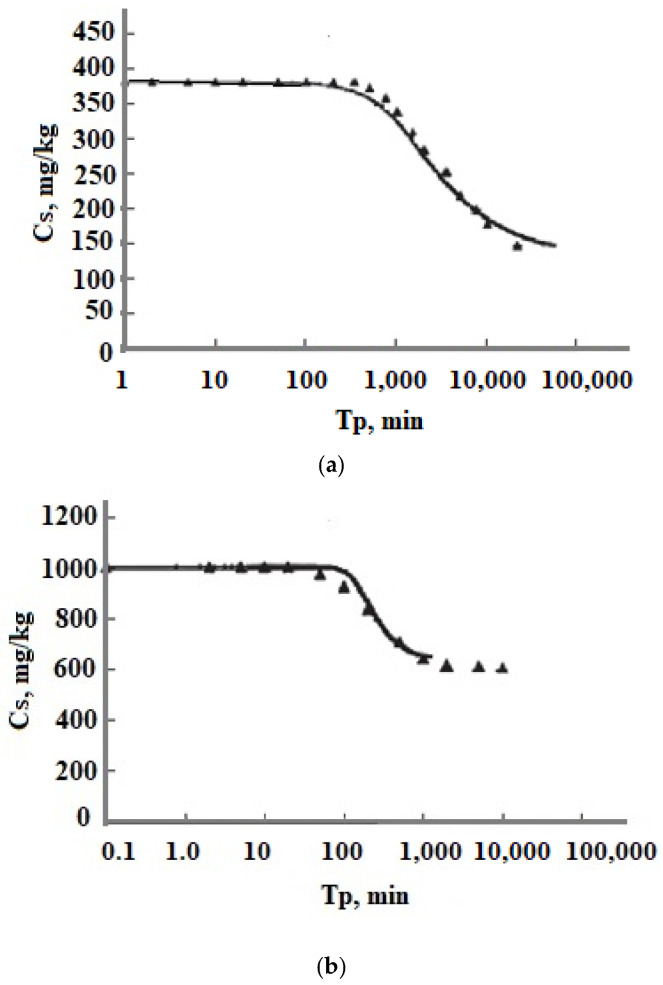
Comparing the experimental data [19] with the results of the numerical simulation. (**a**) t = 30 °C, C_s_ = 383 mg/kg, pH 7.0; (**b**) t = 150 °C, C_s_ = 1008 mg/kg, pH 7.85. I_s_ = 0.000132–0.00134 mol/kg. ▬—results of the simulation; ▲—experimental data.

**Figure 12 polymers-14-04044-f012:**
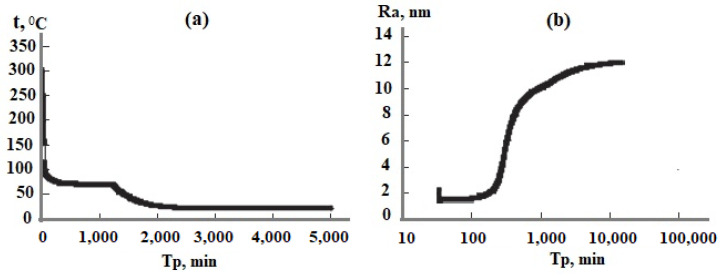
Parameters involved in the technological flow sheet of silica extraction. (**a**) Temperature versus time; (**b**) average silica particle radius as a function of time.

**Figure 13 polymers-14-04044-f013:**
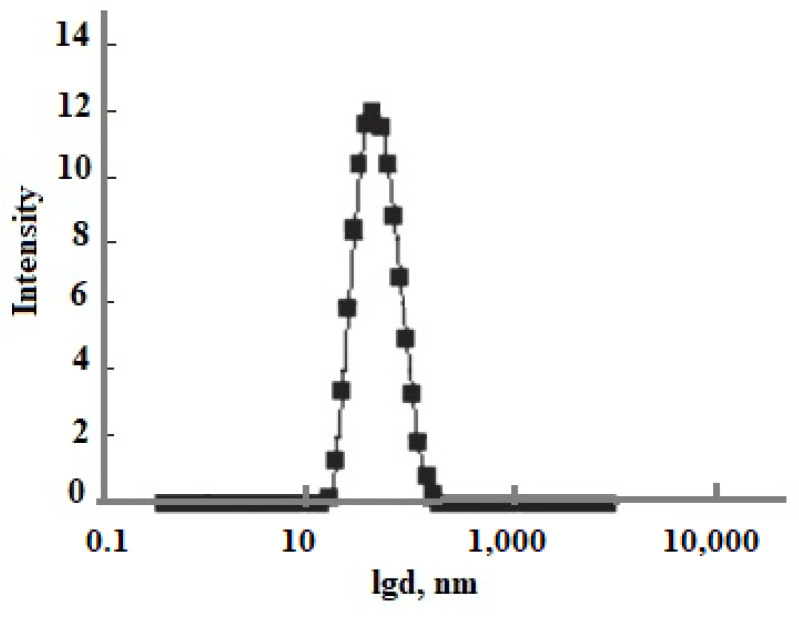
The particle size distribution based on DLS results. d is the particle diameter, nm. Well 054 at the Mutnovsky field.

**Table 1 polymers-14-04044-t001:** Characteristics of the hydrothermal solution from the Mutnovsky field, ionic strength I_s_ = 14.218 mmol/kg, I_s_ = ∑c_j_∙z_j_^2^/2, c_j,_ z_j_ (concentration (mol/kg) and charge of the j-th ion in solution), (z_Na_ = 1, z_K_ = 1, z_Ca_ = 2, z_Mg_ = 2, z_Al_ = 3), pH = 9.35, elec trical conductivity σ*_el_* = 1.1–1.3 mS/cm. nc—not calculated.

Component	Concentration	
	mg/L	mg equ/L
Na^+^	239.4	10.413
K^+^	42.0	1.074
Ca^2+^	1.6	0.0798
Mg^2+^	0.72	0.0592
Fe^2, 3+^	<0.1	<0.0053
Al^3+^	0.27	0.033
NH_4_^+^	1.1	0.0609
Li^+^	0.71	0.102
Total cation concentration	285.9	11.827
Cl^−^	198.5	5.591
HCO_3_^−^	81.0	1.327
CO_3_^2−^	19.9	0.663
SO_4_^2−^	192.1	3.9995
HS^−^	4.95	0.15
H_2_S^0^	5.92	-
F^−^	n/d	n/d
Total anion concentration	496.5	11.73
H_3_BO_3_	106.9	nc
(H_4_SiO_4_)_t_	1190	nc
(H_4_SiO_4_)_s_	222	nc
Mineralization M_h_	1638.9	nc

**Table 2 polymers-14-04044-t002:** Values of function f at different pH.

pH	f(pH)	pH	f(pH)	pH	f(pH)
4.0	0.00025	5.9	0.017	7.5	0.21
4.5	0.00079	6.0	0.022	8.0	0.32
5.0	0.0024	6.5	0.055	8.5	0.44
5.5	0.0075	7.0	0.119	8.7	0.49

**Table 3 polymers-14-04044-t003:** Values of function i at different pH.

pH	i (pH)
6.0	0.01
6.5	0.03
7.0	0.07
7.5	0.15
8.0	0.29
8.5	0.48
8.7	0.56

**Table 4 polymers-14-04044-t004:** Dependence of Zeldovich factor Z on the temperature and pH (-).

**t, °C**	**pH**					
	**4.0**	**4.5**	**5.0**	**5.5**	**6.0**	**6.5**
20	3.8∙10^−2^	3.8∙10^−2^	3.8∙10^−2^	3.8∙10^−2^	3.8∙10^−2^	3.9∙10^−2^
50	2.1∙10^−2^	2.1∙10^−2^	2.1∙10^−2^	2.1∙10^−2^	2.2∙10^−2^	2.2∙10^−2^
100	5.8∙10^−3^	-	5.8∙10^−3^	5.8∙10^−3^	6.0∙10^−3^	6.1∙10^−3^
130	1.4∙10^−3^	1.3∙10^−3^	1.3∙10^−3^	1.3∙10^−3^	1.4∙10^−3^	1.4∙10^−3^
**t, °C**	**pH**					
	**7.0**	**7.5**	**8.0**	**8.5**	**8.7**	**8.99**
20	4.2∙10^−2^	4.8∙10^−2^	6.4∙10^−2^	1.1∙10^−1^	1.5∙10^−1^	3.3∙10^−1^
50	2.4∙10^−2^	2.8∙10^−2^	4∙10^−2^	7.6∙10^−2^	1.2∙10^−1^	3.5∙10^−1^
100	6.7∙10^−3^	8.2∙10^−3^	1.2∙10^−2^	3.1∙10^−2^	6.2∙10^−2^	8.6∙10^−3^
130	1.6∙10^−3^	2∙10^−3^	3.3∙10^−3^	1∙10^−2^	2.6∙10^−4^	-

**Table 5 polymers-14-04044-t005:** The final average radius of silica particles at a constant temperature and varying pH (nm). C_s_(t = 0) = C_t_ = 700 mg/kg.

pH	t, °C					
	20	40	50	60	80	100
4	5.28	19.24	48.06	156.55	6342.1	-
5	5.29	19.05	47.30	153.15	5858.7	-
6	2.98	16.82	40.69	128.01	4684.7	-
7	2.97	7.66	15.28	38.17	701.4	-
8	0.98	1.21	1.50	1.77	3.48	18.7
8.5	0.64	0.73	0.78	0.84	1.01	1.26

**Table 6 polymers-14-04044-t006:** Values of the final mean size of silica particles (nm) under different temperatures t and initial concentrations of solution C_s_, pH = 8; (-) means that it was not estimated.

**C_S_,** **mg/kg**	**t, °C**			
	**20**	**40**	**50**	**60**
300	93.5	-	-	-
350	12.4	1019.9	-	-
400	4.35	49.1	586.3	-
500	1.79	4.08	8.96	32.7
600	1.246	1.84	-	3.80
700	0.98	1.21	-	1.77
800	0.87	1.05	-	1.31
1000	0.73	0.83	-	0.95
**C_S_,** **mg/kg**	**t, °C**			
	**70**	**80**	**85**	**90**
300	-	-	-	-
350	-	-	-	-
400	-	-	-	-
500	-	-	-	-
600	7.56	24.14	56.14	169.8
700	-	3.48	-	6.49
800	-	1.81	-	2.30
1000	-	1.10	-	1.21

**Table 7 polymers-14-04044-t007:** Durations of homogeneous t_hom_ and heterogeneous t_hetg_ nucleation at different temperatures, pH, C_s_.

Temperature, °C	pH	Initial OSA Concentration C_s_, mg/kg	t_hom_. min	t_hetg_, min	t_hom_/t_hetg_
20	5	700	80,385.3	4,369,195	0.018398
20	6	700	8255.45	454,617.5	0.018159
20	7	700	749.63	81,119.24	0.009241
20	8	700	50.20	5133.77	0.009779
50	5	700	118,281.1	3,027,579	0.039068
50	6	700	11,123.75	250,812.4	0.044351
50	7	700	715.87	20,328.97	0.035215
50	8	700	13.47	776.68	0.01735
80	5	700	4,352,938	7.15·10^7^	0.0608
80	6	700	373,867.6	5,574,710	0.067065
80	7	700	10,519.41	143,413.7	0.07335
80	8	700	19.635	328.12	0.05984
100	5	700	7.07·10^9^	5.32·10^10^	0.133
100	6	700	4.89·10^8^	3.53·10^9^	0.139
100	7	700	32,828.60	2.19·10^7^	0.15
100	8	700	116.61	638.40	0.182
120	5	700	1.08·10^19^	4.94·10^19^	0.219
120	6	700	3.93·10^17^	1.78·10^18^	0.22
120	7	700	4.71·10^13^	1.87·10^14^	0.252
120	8	700	26,324.92	73,435.34	0.3584

**Table 8 polymers-14-04044-t008:** The final average size of silica particles in solutions with different ionic strength values. t = 20 °C, pH = 8.0, and C_s_ = 700 mg/kg.

I_S_, mol/kg	R_a_, nm	I_S_, mol/kg	R_a_, nm	I_S_, mol/kg	R_a_, nm
0.014	0.99	0.4	1.048	1.4	1.042
0.07	1.029	0.8	1.049	7.1	0.9
0.14	1.036	1.0	1.047	10.6	0.79
0.28	1.044	102	1.045	14.2	0.7

## Data Availability

The data presented in this study are available on request from the corresponding author.

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
