# Peer review of "Silicic Acid Polymerization and SiO2 Nanoparticle Growth in Hydrothermal Solution"

_polymers, 2022, doi:10.3390/polym14194044_

Round 1
Reviewer 1 Report (Previous Reviewer 2)
The authors have addressed my previous concerns. I recommend the publication in the journal.
Author Response
We thank the reviewer for comments made, whauch alloed us to improve our manuscript. All new changes in text are marked by red.

Reviewer 2 Report (New Reviewer)
The manuscript concerns the numerical simulation of orthosilicic acid polymerization and SiO2 nanoparticles formation in hydrothermal solution. The following issues should be addressed:
· The theoretical model needs to be clearly formulated, using a consistent notation throughout the text; symbols are currently not written consistently throughout the text and the figures in terms of font type, font style (inconsistent use of italics), letter case (uppercase or lowercase), subscripts/superscripts, etc. Given the major role of the theoretical formulation in the manuscript, these inconsistencies prevent a proper presentation of the content. For example t and T are interchangeably used to indicate time and temperature, which is extremely confusing and misleading. Given the large number of symbols used, it would be advisable to introduce a list of symbols (physical quantities, physical constants), with a clear explanation of their meaning and units of measurement and a consistent use throughout the text and the figures. Units of measurements should be indicated with their symbol.
· Several figures are of poor quality and difficult to read (see, for instance, Fig. 2). In some cases there is a superfluous repetition of the same data in figures and tables (see, for example, Table 1 and Fig. 1). Some tables are not properly presented (for example, in Table 7, there are free different quantities reported in the same column, while the second column is empty).
· All the acronyms should be explained on their first appearance.
· The introduction should be reformulated in order to be more focused and appear more compelling for the reader. It should include a clear statement of the problem to be solved, a concise description of the relevant references on which the manuscript is based and a coherent illustration of the hypotheses and objectives of the authors. Currently there is a long list of previous works, but no evident connection is established between previous literature and the purpose of the paper and the research strategy adopted to address the question.
· A careful English revision is recommended to eliminate mistakes and make the text more fluent.
Author Response
All notes to reviewer are in the attached file.

Reviewer 3 Report (New Reviewer)
This study focuses on numerical simulation of orthosilicic acid polymerization/SiO2 particle fomation and understanding the parameters controlling it. A detailed background was provided, calculation was shown within the content and experiments were carried out to support the simulation.
However, here are some major issues within this draft:
- The authors provided detailed background information regarding the parameters controlling the OSA polymerization and the different particle sizes. However, too much details were provided within the introduction. Please keep in mind, the introduction is supposed to provide a general information of the research topic and emphasize the importance of this study. Currently, it looks like a short review paper at the background.
- Following on last point, currently the relevant references were not well organized. The different PH, concentration, time, temperature and particle size seems lack a clear logic. I would recommend the authors try to re-organize them into a better structure.
- Most of the figure qualities are so low within the draft. It is almost impossible for readers to read the figure information by themselves. The figure qualities MUST BE IMPROVED such as Figure 2, Figure 5 and etc
- Following on that, the figure formatting is not consistent within the study.
- In Figure 2, the Y-axis scales are different within each plot. This is very misleading.
- Table 1 and Table 3 are redundant.
- Table 7 is extremely hard for readers to understand. Particularly the decision of combining temperature, pH and OSA concentration into a column. Please separate them and make the table easier for readers to understand.
- Figure 8 was not mentioned within this study.
- Figure 10 is very confusing. It has 6 sub plots in total. But the caption was only for 2.
- In Line 453, the claimed Newton’s law of heat transfer seems problematic. This Q there should be the power instead of energy. To calculate the energy, the authors should do an integration. I would have to question the results presented in Figure 11.
- It seems all the experimental data used for simulation is from the literatures, is it true? Please comment on that
- The structure of this draft is not well organized. And it seems very redundant in general. Please reorganize the structure within the paper and make it easier for readers to understand the importance of the work.
- All the plot should be remade and be consistent within the draft. Otherwise I don’t think it qualifies for a publication.
In addition, here are some minor problems within the draft:
- Some abbreviations are not defined before being used.
- The equation included in the draft has poor formatting. Please improve that
- Please check all the formatting within the draft.
Author Response
All notes for the reviewer are in the attached file.

Round 2
Reviewer 3 Report (New Reviewer)
Thank you for the revision. The quality of this paper has improved a lot after 1st revision. However, there are still some problems within the draft:
1) The figure qualities are still very low. They look very blurry. I would suggest the authors try to use "Origin" to replot every figure. Current figure qualities cannot meet the requirements of the journal;
2) The equation formats can still be improved.
Author Response
The authors thanks the reviewer for all comments and suggestions made.
In the revised version all corrections are marked in blue.
In Introduction the equations were improved and marked in blue.
All figures were improved and marked in blue.

Round 3
Reviewer 3 Report (New Reviewer)
Thanks for updating the new version. I do see some changes within the updated draft. However, current version does not meet the requirement of publication.
For example:
- Only Fig. 1 - 3 are updated. For other figures, they are still the same as previous version. Please update all the figures within the draft and make sure the formats are consistent.
- Follow last point, the formats changes all the time within the draft. For example, the size of figures are very different. Those qualities cannot be published.
- For line 547 - 555, it seems the line space is different with others. Please change them.
- The font size within the reference part is different with others.
- For equation formats, they are not consistent either. For example, equation line 337 and 338, they are aligned at the right side but not consistent with others.
- Table formats are not consistent either. For example, Table 2 does not have the right formatting.
- There are some other minor issues within the draft. For example, there are several different formatting for ph description including, “”pH=2”(no space), “pH =2”(space at the left side of equal sign), “pH= 2”(space at the right side of equal sign), “pH = 2”(spaces at the both sides of equal sign).
Without consistent formatting in the draft, I do not feel the draft can be published.
Author Response
Thanks for comments on compliance with the publication requirements. Notes to reviewer are in the attached file.

Round 4
Reviewer 3 Report (New Reviewer)
The authors addressed all my questions. It is good to be published now.
This manuscript is a resubmission of an earlier submission. The following is a list of the peer review reports and author responses from that submission.
Round 1
Reviewer 1 Report
All data, calculation procedures and results obtained are a copy of an early published work!!!!!!
Physical Chemistry | p-ISSN: 2167-7042 e-ISSN: 2167-7069 | 2014; 4(1): 1-10 | doi:10.5923/j.pc.20140401.01
The differences between a peer-reviewed manuscript and a previously published article are
Authors: V., Kashutina I. A., Shunina E. = Potapov V. V. => V. V. Potapov, A A. Cerdan and D. S. Gorev
Title: Numerical Modeling of Orthosilicic Acid Nucleation in Hydrothermal Solutions => Silicic acid polymerization and SiO2 nanoparticles growth in hydrothermal solution (the current title is inadequate to the content of the work)
Introduction: In the manuscript based mainly on the literature from the 50-70s
Summary: In the manuscript this summary is enigmatic.
The remaining parts of the work were copied copy past (only the data in the tables was presented with less precision).
The manuscript cannot be accepted for publication!!!!!
Author Response
The present mauscript have another aim, not geochemical process, but technology of hydrothermal synthesis of SiO2 nanoparticles. The Conclusions were another.
In the final version the results on influence of ionic strength and temperature have been added. The Conclusions have been changed. All changes were marked by red colour in the final version.

Reviewer 2 Report
In this manuscript, the authors investigated the kinetics of orthosilicic acid polymerization and SiO2 nanoparticle formation in hydrothermal solutions through numerical simulation. I would recommend the acceptance of the manuscript after the following revision:
- The introduction section needs to be reorganized. The current introduction focuses on literature review, which is good. However, the novelty and advance of this work should also be highlight and illustrated. In addition, I suggest the authors to give a general introduction about nanoparticle synthesis and emphasize why hydrothermal synthesis was studied in this manuscript. Some papers can be cited about different nanoparticle synthesis methods: Materials Today: Proceedings 2.4-5 (2015): 3575-3579.; Journal of the American Chemical Society 143.7 (2021): 2688-2693.; Chemical Society Reviews 44.16 (2015): 5778-5792.
- Figures 12 and 13, Figure legend should be in English.
- Can the authors add a table in the manuscript to compare the results obtained from the current simulation with other existing work using different simulation method?
Author Response
Thank you for the effort significantly improved manuscript. All changes are marked by red colour.
- The Introduction were cited the most reliable papers on 1) experimantal and 2) modelling study of OSA plymerization. In the final version the new papers were cited on the methods of concentration and superconcentration models of OSA plymerization and on the method of molecular dynamic simulation [33, 34, 35]. The papers on synthesis of silica sol based on precursor TEOS [37] and on grenn synthesis of another particles [36, 38] were added. We have developed the hydrothermal synthesis in many previous papers, for example [25, 26, 27, 28], in which explanation of advantages of hydrothermal synthesis were explained. The technology of hydrothermal nanosilica production exclude chemical reagents. such as Na2SiO3, ion-exchange resins, acids. The production cost of hydrothermal nanosilica is lost due low cost of electricity of geothermal electric power plants. Technology can vary particles diameters and specific surface area by the temperature , pH and ionic strength on the stage of OSA polymerization.
- The legends on the Figures 12, 13 are in English in the final version.
- In the end of the manuscript we added the correspondens with results of modeling study of the work [33], in which another models used, and we present a lot off Figures with results of our experimental and modeling studies, results of modeling and experimantal studies of another authors, not only on kinetics of OSA polymerization and on SiO2 particles diameters too.

Reviewer 3 Report
Authors present a numerical simulation study of OSA polymerization in hydrothermal solution towards nanosilica and its role in processes of mineralization.
Although this manuscript includes a good introduction and high amount of results about the growth mechanism of nanosilica in good correlation with experimental studies reported elsewhere, in my opinion, the manuscript can not be accepted for publication because it duplicates high amount of work already published in:
Potapov, V. V., Cerdan, A. A., & Kashutina, I. A. (2019). Numerical Simulation of the Polycondensation of Orthosilicic Acid and of the Formation of Silica Particles in Hydrothermal Solutions. Journal of Volcanology and Seismology, 13(4), 216–225. doi:10.1134/s0742046319040055.
(Not cited in the present manuscript).
Also, there is not enough contribution to the existent literature.
Other minor issues are: typos errors in the whole manuscript, graphics in figures 2, 3, 5 and 7 uses wrong notation for numbers (e.g. E10-2 and so on), references for experimental data are absent in the caption of each figure (e.g. Figures 8, 9, 10, 12 and 13).
Author Response
The aim of the present work was regulation of SiO2 nanoparticles sizes and concentration in hydrothermal synthesis, not geochemical processes. All chenges made in the final version are marked by red colour.
New pragrafs were added on the influence temperature and ionic strength on kinetics of OSA polymerization and on the particles distribution on diameters.
In introduction new papers were cited on the methods of experimantal and modeling study of OSA plymerization.
All Figures were improved.

Round 2
Reviewer 3 Report
Although the manuscript has been improved in terms of further discussion and data, it still duplicates several data from their previous work omitting its reference:
Potapov, V. V., Cerdan, A. A., & Kashutina, I. A. (2019). Numerical Simulation of the Polycondensation of Orthosilicic Acid and of the Formation of Silica Particles in Hydrothermal Solutions. Journal of Volcanology and Seismology, 13(4), 216–225. doi:10.1134/s0742046319040055.
Particularly, the following data and simulation curves are exactly the same:
Fig.1 (this manuscript) = Fig.8 (doi:10.1134/s0742046319040055)
Fig.2 (this manuscript) = Fig.9 (doi:10.1134/s0742046319040055)
Fig.3 (this manuscript) = Fig.10 (doi:10.1134/s0742046319040055)
Fig.4 (this manuscript) = Fig.11 (doi:10.1134/s0742046319040055)
Fig.5 (this manuscript) = Fig.12 (doi:10.1134/s0742046319040055)
In addition, other minor issues were not amended: graphics in figures 2 & 17 use wrong notation for numbers (e.g. E10-2 and so on) and references for experimental data are absent in the caption of each figure.
For these reasons, I think that the manuscript can not be published in Polymers.